



# Estimation of rate coefficients for the reactions of O₃ with unsaturated organic compounds for use in automated mechanism construction

Michael E. Jenkin[1], Richard Valorso[2], Bernard Aumont[2], Mike J. Newland[3], Andrew R. Rickard[3,4]

[1] Atmospheric Chemistry Services, Okehampton, Devon, EX20 4QB, UK
[2] LISA, UMR CNRS 7583, Université Paris-Est Créteil, Université de Paris, Institut Pierre Simon Laplace (IPSL), Créteil, France
[3] Wolfson Atmospheric Chemistry Laboratories, Department of Chemistry, University of York, York, YO10 5DD, UK
[4] National Centre for Atmospheric Science, University of York, York, YO10 5DD, UK

*Correspondence to*: Michael E. Jenkin (atmos.chem@btinternet.com)

**Abstract.** Reaction with ozone ($O_3$) is an important removal process for unsaturated volatile organic compounds (VOCs) in the atmosphere. Rate coefficients for reactions of $O_3$ with VOCs are therefore essential parameters for chemical mechanisms used in chemistry transport models. Updated and extended structure–activity relationship (SAR) methods are presented for the reactions of $O_3$ with mono- and poly-unsaturated organic compounds. The methods are optimized using a preferred set of data including reactions of $O_3$ with 222 unsaturated compounds. For conjugated dialkene structures, site specific rates are defined, and for isolated poly-alkenes rates are defined for each double bond to determine the branching ratios for primary ozonide formation. The information can therefore guide the representation of the $O_3$ reactions in the next generation of explicit detailed chemical mechanisms.

## 1 Introduction

Volatile Organic Compounds (VOCs) are emitted to the atmosphere from both biogenic and anthropogenic sources. Many of these compounds are unsaturated (i.e. contain at least one double bond), including the ubiquitous biogenic VOCs isoprene and monoterpenes (Sindelarova et al., 2014). Chemical degradation of these compounds in the atmosphere leads to a variety of secondary pollutants including ozone and secondary organic aerosol (SOA). Unsaturated compounds are generally highly reactive, and react with the oxidant ozone ($O_3$), which is typically present in the troposphere at mixing ratios in the range 10 – 200 ppb. The ozonolysis reaction involves the concerted addition of $O_3$ to the double bond, followed by decomposition of the short-lived primary ozonide to yield a carbonyl compound and a carbonyl oxide, commonly referred to as a Criegee intermediate (Criegee, 1975). The reaction is important as a non-photolytic source of radicals and reactive intermediates, including the hydroxyl radical (e.g. Johnson and Marston, 2008; Cox et al., 2020). Ozonolysis of large alkenes (e.g. monoterpenes and sesquiterpenes) is also particularly efficient at producing SOA (Hallquist et al., 2009), including as a result of the formation of low volatility products from reactions of Criegee intermediates with atmospheric trace gases (e.g.





Heaton et al., 2007; Sakamoto et al., 2013; Zhao et al., 2015; Mackenzie-Rae et al., 2018; Chhantyal-Pun et al., 2018), and from auto-oxidation mechanisms involving peroxy radicals formed from decomposition of the Criegee intermediates (e.g. Ehn et al., 2014; Jokinen et al., 2015).

Previous assessments using explicit organic degradation mechanisms have demonstrated that the atmosphere contains an almost limitless number of organic compounds (e.g. Aumont et al., 2005), for which it is impractical to carry out experimental kinetics studies. This has resulted in the development of estimation methods for rate coefficients (e.g. see
Calvert et al., 2000; 2011; McGillen et al., 2008; Vereecken et al., 2018; and references therein), which have been applied widely in chemical mechanisms and impact assessments. As part of the present work, a set of preferred kinetic data has been assembled for the reactions of $O_3$ with 222 unsaturated organic compounds, based on reported experimental studies (see Section 2 for further details). Updated structure–activity relationship (SAR) methods are presented for the initial reactions of $O_3$ with unsaturated organic compounds. In the cases of poly-alkenes, the rate coefficient is defined in terms of a summation
of partial rate coefficients for $O_3$ reaction at each relevant site in the given organic compound, so that the attack distribution is also defined. Application of the methods is illustrated with examples in the Supplement.

The information is currently being used to guide the representation of the $O_3$-initiation reactions in the next generation of explicit detailed chemical mechanisms, based on the Generator for Explicit Chemistry and Kinetics of Organics in the Atmosphere (GECKO-A; Aumont et al., 2005), and the Master Chemical Mechanism (MCM; Saunders et al., 2003). It
therefore contributes to a revised and updated set of rules that can be used in automated mechanism construction, and provides formal documentation of the methods. This paper is part of a series of publications, including rules for the estimation of rate coefficients and branching ratios for the reactions of OH with aliphatic (Jenkin et al., 2018a) and aromatic (Jenkin et al., 2018b) organic compounds, and for peroxy radical reactions (Jenkin et al., 2019). Rules governing the decomposition of the primary ozonides, formed initially from the $O_3$-initiation reactions, and the subsequent chemistry of the
Criegee intermediates formed, are considered in a further paper (Mouchel-Vallon et al., 2020).

## 2 Preferred kinetic data

A set of preferred kinetic data has been assembled from which to develop and validate the estimation methods for the $O_3$ rate coefficients. The complete set includes 298 K data for 222 compounds, comprising 111 alkenes and 111 unsaturated oxygenated compounds. Temperature dependences are also defined for a subset of 39 compounds. In three cases, the
preferred rate coefficient is an upper limit value, and in three cases a lower limit value. The information is provided as a part of the Supplement (spreadsheets SI_1 and SI_2). As described in more detail in Section 4, the oxygenates include both monofunctional and multifunctional compounds containing a variety of functional groups that are prevalent in both emitted VOCs and their degradation products, namely -OH, -C(=O)H, -C(=O)-, -O-, -C(=O)OH, -C(=O)O-, -OC(=O)-, -$ONO_2$ and -C(=O)$OONO_2$. For a core set of 30 reactions, preferred kinetic data are based on the evaluations of the IUPAC Task Group
on Atmospheric Chemical Kinetic Data Evaluation (Cox et al., 2020: http://iupac.pole-ether.fr/). The remaining values are





informed by recommendations from other key evaluations with complementary coverage (e.g. Atkinson and Arey, 2003; Calvert et al., 2011; 2015), and have been revised and expanded following review and evaluation of additional data not included in those studies (as identified in spreadsheets SI_1 and SI_2).

## 3 Alkenes

As discussed in detail previously (e.g. Calvert et al., 2000; Vereecken et al., 2018 and references therein), the data indicate that the rate coefficients are highly sensitive to alkene structure, and depend on the degree of alkyl substitution of the unsaturated bond(s), on steric effects, and on ring-strain effects in cyclic compounds. The set of preferred kinetic data has been used to update and extend a structure-activity relationship (SAR) method that can be used to estimate the rate coefficients when no experimental determinations are available. Similar to previous appraisals (e.g. Jenkin et al., 1997;

Calvert et al., 2000), reference rate coefficients ($k$) are defined for addition of $O_3$ to a series of alkene and conjugated dialkene structures, based on the preferred data for relevant sets of alkenes and conjugated dialkenes.

### 3.1 Acyclic monoalkenes

The set of preferred values contains data for the reactions of $O_3$ with 43 acyclic monoalkenes. The generic rate coefficients for $O_3$ addition to C=C bonds in acyclic monoalkene structures with differing extents of alkyl (R) substitution are given in

Table 1 ($k_{A1O3}$-$k_{A6O3}$). These rate coefficients are based on averages of the preferred values for the identified sets of alkenes, and are defined for "R" being a linear alkyl group (i.e. -$CH_3$ or -$CH_2R'$). In practice, reported data for sets of alk-1-enes ($CH_2$=CHR) and 2-methylalk-1-enes (the majority of the $CH_2$=$CR_2$ dataset) show small systematic increases in $k$ with the size of the alkyl group (Mason et al., 2009), although the preferred data for the other structural alkene groups do not apparently show such dependences (see Fig. 1). In view of the high sensitivity of $k$ to alkene structure (as indicated above),

the use of single size-independent values of $k$ for each of these structural groups is considered acceptable for the present SAR.

Reported rate coefficients for monoalkenes possessing branched alkyl groups tend to be lower than those for alkenes possessing the corresponding linear alkyl groups, as is generally apparent from the data in Fig. 1. In most reported cases, branching occurs at the carbon atom adjacent ($\alpha$) to the alkene structure, which may influence $O_3$ attack through steric

hindrance (e.g. Calvert et al., 2000; Johnson et al., 2000). Rate coefficients for alkenes with substituents at the $\alpha$ position are determined from the following expression,

$$k_{calc} = k_{AO3} \, \Pi F_\alpha(X) \tag{1}$$

where $k_{AO3}$ is the appropriate reference rate coefficient in Table 1, and a value of $F_\alpha(X)$ is applied for each $\alpha$ substituent in the molecule. A factor, $F_\alpha$(alkyl), describing the effect of each (acyclic) alkyl group at the $\alpha$ carbon atom was determined, by

minimizing the summed square deviation, $\Sigma((k_{calc}-k_{obs})/k_{obs})^2$ for the set of relevant branched alkenes (the resultant value is given in Table 2, along with those for selected oxygenated groups discussed below). It should be noted that the reported



value of $k_{obs}$ for 3,4-diethyl-hex-2-ene is substantially lower than the reference value of $k_{A5O3}$ for the CHR=CR$_2$ structure (i.e. by two orders of magnitude; see Table 1), and this compound was therefore excluded from the optimization procedure for $F_\alpha$(alkyl). Confirmatory measurements of that rate coefficient, and data for other α-branched alkenes, are therefore
required to test and refine the method proposed here. The limited data for more remotely branched alkenes suggest no significant effect on the rate coefficient. In those cases, the rate coefficients in Table 1 are applied as a default.

The corresponding absolute deviations, $(k_{calc}-k_{obs})/k_{obs}$, at 298 K for 40 acyclic monoalkenes in the set of preferred values indicate that the estimation method reproduces the observed values to within about $^{+50\%}_{-30\%}$ (see also Fig. 2). The three monoalkenes excluded from the procedure were ethene (a unique structure, for which no value needs to be calculated), 3,4-
diethyl-hex-2-ene (as indicated above) and 3,4-dimethyl-hex-3-ene, for which only a lower limit preferred value is available. In the final case, $k_{calc}$ is a factor of three higher than the lower limit value.

### 3.2 Cyclic monoalkenes

The set of preferred values contains data for reactions of O$_3$ with 14 simple monocyclic monoalkenes containing endocyclic double bonds, including cyclopentenes, cyclohexenes, cycloheptenes, cyclooctenes and cyclodecenes. The values of $k$ for
these sets of compounds show systematic deviations from those observed for acyclic monoalkenes with the same level of substitution, likely resulting from the effects of ring-strain (e.g. Calvert et al., 2000). Table 3 provides a series of ring factors at 298 K, $F_{ring}$, based on optimization to this dataset. The rate coefficients for cyclic alkenes with endocyclic double bonds are therefore determined from the following expression,

$$k_{calc} = k_{AO3} \prod F_{ring} F_\alpha(X) \qquad (2)$$

where $k_{AO3}$ is the appropriate reference rate coefficient ($k_{A3O3}$-$k_{A6O3}$) in Table 1. For polycyclic alkenes, a value of $F_{ring}$ needs to be applied for each ring for which the given C=C bond is a component. In addition to the $F_{ring}$ values given in Table 3, it is also possible to infer a tentative value of 12 for $F_{ring\,(298)}$ for 11-member rings, based on the reported rate coefficient for the sesquiterpene α-humulene (which contains a 1,4,8-cycloundecatriene ring), with the assumption that the values for $F_{ring}$ can be applied to cyclic systems with unconjugated multiple double bonds. The reported rate coefficient for the sesquiterpene β-
caryophyllene (which contains a *trans*-cyclononene ring) then allows a tentative value of $F_{ring\,(298)} = 2.1$ for 9-member rings, although it is noted that the level of ring-strain, and therefore $F_{ring}$, likely depends on the *cis-/trans-* conformation. Clearly, additional data for cyclononenes and cycloundecenes are required to confirm these tentative values of $F_{ring}$.

As with the acyclic alkenes above, a value of $F_\alpha$(alkyl) is also applied for each (acyclic) alkyl group at the α carbon atom in both monocyclic and polycyclic alkenes, where appropriate. For this procedure, the term "acyclic" is taken to mean that the
first carbon atom in the substituent group is not part of a cycle. To avoid ambiguity in defining the number of α acyclic alkyl groups, the base structure is taken to be cyclic as a default, and values of $F_\alpha$(alkyl) are applied as appropriate to each acyclic alkyl group. This rule applies whether the double bond is endocyclic or exocyclic, and has the effect of maximizing the number of acyclic alkyl groups (e.g. see example calculations B2–B5 in the Supplement).





### 3.3 Acyclic conjugated dialkenes

The generic rate coefficients for $O_3$ addition to C=C-C=C bond structures in acyclic conjugated dialkenes with differing extents of alkyl (R) substitution are given in Table 4, based on reported data for 10 compounds ($k_{D1O3}$-$k_{D11O3}$). These rate coefficients are based on the preferred values of $k$ for the identified dialkenes, with those for some structural groups being inferred from the observed trends in the impact of successive alkyl substitution on $k$. The values are generally based on data for conjugated dialkenes for which "R" is a linear alkyl group (these making up almost all of the reported data). The limited information on dialkenes possessing branched substituent groups (5-methyl-hexa-1,3-diene and 5,5-dimethylhexa-1,3-diene) suggests that there is a less-pronounced reducing effect on $k$, compared with that observed for the monoalkenes above. Similar to the approach used for the reactions of OH with conjugated dialkenes (Jenkin et al., 2018a), the following expression is therefore applied,

$$k_{calc} = k_{DO3} \prod (F_\alpha(X))^{\frac{1}{2}} \tag{3}$$

where $k_{DO3}$ is the appropriate reference rate coefficient in Table 4, and a value of $F_\alpha(X)$ (Table 2) is applied for each $\alpha$ substituent in the molecule.

### 3.4 Cyclic conjugated dialkenes

The set of preferred values contains data for the reactions of $O_3$ with five monocyclic conjugated dialkenes, including cyclohexa-1,3-dienes, cyclohepta-1,3-diene and cycloocta-1,3-diene. The values of $k$ for these sets of compounds all show systematic deviations from those observed for acyclic conjugated dialkenes with the same level of substitution, again likely resulting from the effects of ring-strain (e.g. Calvert et al., 2000; Lewin et al., 2001). In each case, these also differ substantially from those for the same sized cyclic monoalkenes (as shown in Table 3), and Table 5 shows a series of ring factors at 298 K, $F'_{ring}$, based on optimization to the cyclic conjugated dialkene dataset. The rate coefficients for cyclic conjugated dialkenes are therefore determined from the following expression:

$$k_{calc} = k_{DO3} \prod F'_{ring} F_\alpha(X)^{\frac{1}{2}} \tag{4}$$

For polycyclic systems, a value of $F'_{ring}$ needs to be applied for each ring for which the given C=C-C=C bond structure is a component. As for acyclic conjugated dialkenes, a value of $F_\alpha(X)^{\frac{1}{2}}$ is applied for each $\alpha$ substituent in the molecule, where appropriate. In the cases of cyclic (or polycyclic) conjugated dialkenes with $\alpha$ acyclic alkyl substituents, the base structure is once again taken to be cyclic, and values of $F_\alpha(alkyl)$ are applied as appropriate (e.g. see example calculations D1 and D2 in the Supplement). Note that for the special case of conjugated dialkenes for which only one of the double bonds is within the ring (e.g. β-phellandrene: example D2 in the Supplement), a modified version of Eq. (4) is applied, in which $F'_{ring}$ is replaced by the appropriate value of $F_{ring}$.





### 3.5 Other alkenes and poly-alkenes

The remainder of the alkene dataset consists of preferred values for 30 acyclic and cyclic compounds containing various combinations of isolated double bonds and conjugated dialkene structures, for which the methods described above can be used to estimate rate coefficients. There are also preferred values for a limited set of three conjugated poly-alkenes (one acyclic and two cyclic) and four alk-1-enyl-substituted aromatics (styrenes), for which there are insufficient data to attempt development of a SAR method.

The observed and calculated rate coefficients for 100 alkenes and poly-alkenes are compared in the correlation plot in Fig. 2. These are all the compounds for which preferred values are available in the reference database, less those not covered by the SAR methods, i.e. the three conjugated poly-alkenes and four styrenes referred to above; and ethene and buta-1,3-diene which are unique structures. As shown in Fig. 2, the SAR methods perform well for the sets of acyclic monoalkenes and conjugated dialkenes, and for the monocyclic monoalkenes and conjugated dialkenes. This is because the data show well

defined variations with structure, and because many of those data were used as the basis of the SAR methods. The data for the remaining 30 compounds are subdivided into acyclic, monocyclic and polycyclic in Fig. 2, with the observed data from the first two categories also generally well described by the SAR methods. The observed values of $k$ for the remaining polycyclic compounds are also reasonably well correlated, although with much more scatter than for the simpler structures. This is almost certainly due to a combination of ring-strain and steric effects in these complex structures that cannot be fully

accounted for by the SAR methods developed here.

### 4 Unsaturated oxygenated compounds

The set of preferred values contains data for the reactions of $O_3$ with 111 unsaturated oxygenated compounds, possessing -OH, -C(=O)H, -C(=O)-, -O-, -C(=O)OH, -C(=O)O-, -OC(=O)-, -ONO$_2$ and -C(=O)OONO$_2$ substituents. The SAR methods applied to these compounds depend on the location of the substituent oxygenated group relative to the C=C bond, and fall

into three categories. For those possessing oxygenated substituents at the α position (i.e. α,β-unsaturated, or allylic, oxygenates), the methods described above for alkenes and dialkenes are modified to take account of the effect of the given substituent (see Sect. 4.1). More remote oxygenated substituents are assumed to have no effect, and the appropriate alkene or dialkene rate coefficient is applied unmodified in these cases (see Sect. 4.2). When the oxygenated group (including -C(=O)H, -C(=O)- and -C(=O)O-) is a substituent of the C=C group itself (i.e. vinylic oxygenates), the method is based on a

series of reference rate coefficients for those specific structures, which are derived from the preferred data for the relevant sets of oxygenated compound (see Sect. 4.3).





### 4.1 α,β-unsaturated compounds

Preferred kinetics data at 298 K are available for 25 α,β-unsaturated, or allylic, oxygenates, containing -OH (14 compounds), -C(=O)H (3 compounds), -C(=O)R (1 compound), -OR (4 compounds, one possessing a remote -OH group), -OC(=O)R (1 compound), and both -ONO$_2$ and -OH (2 compounds, with one having only a lower limit recommendation). These data were used, in conjunction with the methods described for alkenes and dialkenes in Sect. 3, to optimize the corresponding values of $F_\alpha(X)$ given in Table 2. It was found that the effect of the -C(=O)H and -C(=O)R could reasonably be described by a single factor, $F_\alpha$(-C(=O)-), and the further assumption was made that the same factor applies to groups containing the -C(=O)O- sub-structure. It is noted that several of the factors are based on data for limited sets of compounds (in some cases a single compound), and further data are clearly required to test the approach fully. As shown in Fig. 3, however, the method appears to work very well for most of the relevant compounds containing -OH groups (the largest subset of α,β-unsaturated oxygenates), providing some support for the approach.

### 4.2 Unsaturated compounds containing remote oxygenated substituents

The preferred data also include rate coefficients for 21 unsaturated compounds possessing remote oxygenated substituents. In these cases, the oxygenated substituent is assumed to have no effect, and the corresponding alkene or dialkene rate coefficient, calculated as described in Sect. 3, is applied unmodified. As shown in Fig. 4, this assumption provides 298 K values of $k_{calc}$ that are generally within about a factor of two of the values of $k_{obs}$, and therefore within the scatter of the methods when applied to unsubstituted alkenes and dialkenes.

In the majority of cases, the presence of the remote oxygenated group appears to reduce the value of the rate coefficient slightly, compared with that of the generic alkene or dialkene rate coefficient. In the cases of a series of *cis*-hex-3-enyl esters (i.e. *cis*-CH$_3$CH$_2$CH=CHCH$_2$OC(=O)R) the rate coefficient is reported to depend systematically on the size of the remote R group, rather than displaying a consistent influence of the -OC(=O)- substructure itself (Zhang et al., 2018). Within this series, the rate coefficient for the largest compound (with R = *n*-C$_3$H$_7$) agrees well with the reference rate coefficient, whereas that for the smallest (with R = H) is about a factor of three lower than the reference rate coefficient. Clearly, further information is required for unsaturated compounds possessing remote oxygenated substituents before refined estimation methods can be developed that take account of this type of effect.

### 4.3 Vinylic oxygenated compounds

When the oxygenated group (including -C(=O)H, -C(=O)- and -C(=O)O-) is a substituent of the C=C group itself (i.e. vinylic oxygenates), the data indicate that the rate coefficients are much less sensitive to the presence of other alkyl groups attached to the C=C group (and in some cases actually decrease upon additional substitution). In contrast, the data for some classes clearly show a greater influence of substituent size. It is therefore not possible to treat these compounds using modifications to the SAR methods presented for alkenes in Sect. 3, and it is necessary to assign generic rate coefficients for





addition of $O_3$ to a series of vinylic oxygenate structures. Three categories of vinylic oxygenate structure are considered, namely vinyl aldehydes and ketones (Table 6), vinyl esters and acids (Table 7) and vinyl ethers (Table 8).

The influence of substituent group size is clearly apparent in the data for vinyl aldehydes and ketones, vinyl ethers and alk-1-enoic acid alkyl esters, e.g. as discussed recently by Ren et al. (2019) for the esters. Accordingly, the following expression is used to describe the 298 K data,

$$k_{calc} = k°_{298\ K} \times [1 + \Sigma(\alpha_s(n_i\text{-}1))] \qquad (5)$$

where $k°_{298\ K}$ and $\alpha_s$ are constants. $n_i$ is the number of carbon atoms in the i$^{th}$ substituent group, where each relevant

substituent group is represented by "R" in the structures shown in Tables 6 – 8. $k°_{298\ K}$ therefore quantifies the rate coefficient when each R group in the given structure is $CH_3$, and $\alpha_s \times k°_{298\ K}$ the incremental increase for each additional carbon in any substituent. As defined, therefore, the same incremental increase is assumed to apply to each R group in the given structure, although the trends in the preferred data are generally based on information for particular R groups. Additional data are therefore required to test this assumption. It was also found that there was only marginal benefit in using

independent values of $\alpha_s$ for the different vinylic oxygenate categories, based on the data currently available. A single, category-independent value of $\alpha_s = 0.19$ was therefore optimized for simplicity. The calculated and observed rate coefficients are compared on the scatter plot in Fig. 5.

There is only limited information available on the temperature dependences of these reactions. Where data are available (e.g. for methacrolein and methyl vinyl ketone), the data suggest that the temperature dependence can reasonably be represented

by $k = A \times \exp(\text{-}(E/R)/T)$, with $A = 10^{-15}$ cm$^3$ molecule$^{-1}$ s$^{-1}$, and $E/R = 298 \times \ln(A/k_{calc\ (298\ K)})$, and this approach is adopted in the present work.

There are very limited data for conjugated dialkenes containing vinylic oxygenated substituents and for cyclic vinylic oxygenates, and it is not possible to propose SAR methods for most oxygenated groups at the present time. The data include rate coefficients for some relevant conjugated dienals/dienones and cyclic vinyl ketones (hexa-2,4-diendial, cyclohex-2-en-1-

one, β-ionone and acetyl-cedrene). In contrast to the compounds discussed above, the rate coefficients for these species are all reasonably well described by applying the appropriate conjugated dialkene or alkene rate coefficient in Tables 1 and 4, reduced by a factor of 50 for each alkyl group replaced by a -C(=O)H or -C(=O)- group. This assumption is provisionally applied in the current work, although further data are clearly required. There are also limited data for some furans and dihydrofurans. The rate coefficients for these species are influenced by the compounds being aromatic (in the cases of the

furans) and also by ring-strain effects, and it is difficult to extend the methods developed here for unsaturated ethers to cover these species. The methods presented here are therefore not applicable to heterocyclic compounds with endocyclic double bonds, such as furans and dihydrofurans.

There are no data for a number of vinylic oxygenated functional groups (e.g., -ONO$_2$, -OOH), although such compounds are not expected to be prevalent in atmospheric chemistry. Data for vinylic alcohols are limited to the multifunctional


compound, 3,4-dihydroxy-3-hexene-2,5-dione, and suggest that the presence of hydroxy groups has a limited effect on the rate coefficient. Studies of related compounds tend to be complicated by keto/enol tautomerism.

## 4.4 Combinations of groups

Data for compounds containing two different vinylic oxygenated substituents listed in Tables 6 – 8 are limited to 4-methoxy-but-3-ene-2-one. This therefore falls into both of the CHR=CHC(=O)R and ROCH=CHR generic structure categories, and

the rate coefficients for these categories, $k_{VC7O3}$ and $k_{VO3O3}$, differ by an order of magnitude. The rate coefficient for 4-methoxy-but-3-ene-2-one ($1.3 \times 10^{-17}$ cm$^3$ molecule$^{-1}$ s$^{-1}$) is actually a factor of three lower than that for the less reactive category ($k_{VC7O3}$). Based on this, it is tentatively suggested that the estimated rate coefficient for compounds containing two different vinylic oxygenated substituents should be based on the less reactive category.

Data for compounds containing both vinylic and allylic oxygenated groups are limited to 2-methyl-4-nitrooxy-*cis*-2-buten-1-

al, which contains a vinyl -C(=O)H group and an allyl -ONO$_2$ group. In this case, the rate coefficient ($4.4 \times 10^{-18}$ cm$^3$ molecule$^{-1}$ s$^{-1}$) is in reasonable agreement with that of the relevant vinylic category, $k_{VC4O3}$ (Table 6), suggesting that the allyl -ONO$_2$ group has almost no additional deactivating effect in this case. This is consistent with the relative insensitivity of the vinylic rate coefficients to the presence of additional substituents, and it is therefore tentatively suggested that the appropriate rate coefficient in Tables 6 – 8 can be applied, with no additional effect from an allylic substituent in relevant

cases (i.e. the factors in Table 2 are only applied with rate coefficients derived from those shown in Tables 1 and 4).

## 5 Initial products and branching ratios

It is well established that the addition of O$_3$ to a C=C bond leads to initial formation of a primary ozonide (POZ), or 1,2,3-trioxolane product (e.g. Calvert et al., 2000; Johnson and Marston, 2008). In compounds with multiple C=C bonds, the SAR methods developed here define $k_{calc}$ in terms of a summation of the various alkene and/or conjugated dialkene structures

within the poly-unsaturated compound, and therefore also provide the basis for estimating branching ratios for the formation of the various isomeric POZs (e.g. see calculations B4, B5 and C2 in the Supplement). Using β-ocimene as an example, the present SARs provide a value of $k_{calc} = k_{A5O3} + k_{D4O3} = 5.5 \times 10^{-16}$ cm$^3$ molecule$^{-1}$ s$^{-1}$ at 298 K, which agrees well with the preferred value of $k_{obs} = 5.1 \times 10^{-16}$ cm$^3$ molecule$^{-1}$ s$^{-1}$. The component rate coefficients also indicate that the reaction is expected to occur predominantly (85 %) at the isolated C=C bond, leading to the formation of POZ1, as shown in the

schematic in Fig. 6. This conclusion is also supported by comparison of $k_{obs}$ with that reported for the reaction of O$_3$ with the β-ocimene oxidation product 4-methyl-hexa-3,5-dienal, which retains the conjugated dialkene structure (Baker et al., 2004).

The addition of O$_3$ to conjugated dialkene structures leads to the formation of either of two primary ozonides, as shown in Fig. 6 for the example of the minor channel of β-ocimene ozonolysis (POZ2 and POZ3). In the cases of symmetrically-substituted conjugated dialkene structures (i.e. CH$_2$=C(R)C(R)=CH$_2$, CHR=CHCH=CHR, CR$_2$=CHCH=CR$_2$ and

CHR=C(R)C(R)=CHR, where "R" represents any alkyl group or remotely-substituted oxygenated group) it is reasonable to





assume that the addition of $O_3$ occurs equally at the two possible sites. There has only been limited information reported on the products of the reactions of $O_3$ with unsymmetrically-substituted conjugated dialkenes. Most of this information relates to the reaction of $O_3$ with isoprene ($CH_2=C(CH_3)CH=CH_2$), but with selected product yields reported for subsequently-formed carbonyl compounds in a few other cases (Lewin et al., 2001). In the case of isoprene, product information indicates

that the addition of $O_3$ occurs significantly at both sites, but with about 60 % at the less-substituted $-CH=CH_2$ bond (e.g. Atkinson and Aschmann, 1994; Nguyen et al., 2016), and the same branching ratio can therefore reasonably be assigned to $CH_2=C(R)CH=CH_2$ structures in general. The information for other conjugated dialkenes also suggests preferential addition of $O_3$ at the less-substituted C=C bond (Lewin et al., 2001; Mackenzie-Rae et al., 2016). On this basis, it is tentatively assumed that addition occurs 60 % at a less-substituted C=C that contains one fewer alkyl substituents (e.g. as in

$CH_2=CHCH=CHR$ or $CHR=CHC(R)=CHR$), 70 % at a less-substituted C=C bond that contains two fewer alkyl substituents (e.g. as in $CH_2=CHC(R)=CHR$ or $CHR=CHC(R)=CR_2$) and 80 % at a less-substituted C=C bond that contains three fewer alkyl substituents (i.e. as in $CH_2=CHC(R)=CR_2$ alone). Clearly, further information is required to allow these addition ratios to be assigned with greater certainty. In the absence of reported mechanistic data, the same rules are also applied to conjugated dialkene structures with vinylic or allylic oxygenated groups.

## 295  6 Conclusions

Updated and extended structure–activity relationship (SAR) methods have been developed to estimate rate coefficients for the reactions of $O_3$ with unsaturated organic species. The group contribution methods were optimized using a database including a set of preferred rate coefficients for 222 species. The overall performance of the SARs in determining $\log k_{298\ K}$ is now summarized.

The distribution of errors ($\log k_{calc}/k_{obs}$), the root mean squared error (RMSE), the mean absolute error (MAE) and the mean bias error (MBE) were examined to assess the overall reliability of the SAR. The RMSE, MAE and MBE are here defined as:

$$RMSE = \sqrt{\frac{1}{n}\sum_{i=1}^{n}(\log k_{calc} - \log k_{obs})^2} \qquad (20)$$

$$MAE = \frac{1}{n}\sum_{i=1}^{n}|\log k_{calc} - \log k_{obs}| \qquad (21)$$

$$MBE = \frac{1}{n}\sum_{i=1}^{n}(\log k_{calc} - \log k_{obs}) \qquad (22)$$

where $n$ is the number of species in the dataset. A total of 199 of the 222 species in the database contributed to the statistical analysis. Six species could not be included, because only upper or lower limit recommendations are available. In addition, the SAR methods do not currently include styrenes, heterocyclic species and conjugated poly-alkenes (11 species); and the smallest species in some homologous series (i.e. ethene, buta-1,3-diene, acrolein, butenedial and acrylic acid) are not



covered by the SAR categories because the double bonds do not contain (additional) organic substituents (see Sects. 3 and 4). Finally, because of the factor of 60 difference between $k_{calc}$ and $k_{obs}$, 3,4-diethylhex-2-ene was also excluded from the statistical analysis as an outlier (see Sect. 3.1). However, it is emphasised that there is no firm basis for believing that the reported rate coefficient for 3,4-diethylhex-2-ene (Grosjean and Grosjean, 1996) is any less reliable than many other rate coefficients in the database. Given the substantial disagreement in $k_{calc}$ and $k_{obs}$, confirmatory measurements of that rate coefficient, and data for other similar branched alkenes, would clearly be valuable to help test and refine the methods presented here.

Figure 7 summarises the statistics for the full set of 199 species, for acyclic and cyclic species collectively, and for various alkene and unsaturated oxygenate subsets. With the exception of the poly-alkene and remotely-substituted oxygenate subsets, the calculated log $k_{298\,K}$ for all categories shows no significant bias, with MBE remaining below 0.07 log units, and with median values of the error distributions close to zero. Overall, the SAR methods overestimate $k_{298\,K}$ for poly-alkenes and remotely-substituted oxygenates by a factor of about 1.5. This is likely due to a number of contributory factors that are not fully accounted for in the SAR methods, including effects of remote substituents on double bond reactivity (e.g. see Sect. 4.2) and possible systematic ring strain effects in cyclic poly-alkenes that are incompatible with the factors derived from simpler compounds (see Sects. 3.2 and 3.4).

The RMSE for the various alkene and unsaturated oxygenate subsets cover the range from 0.11 to 0.27 log units, i.e. the relative errors for the calculated $k_{298\,K}$ lie in the range 29 % − 86 %. Of these, the poly-alkene and remotely-substituted oxygenate subsets again have values towards the high end of the range (0.27 and 0.22, respectively). The RMSE for the mono-alkene subset is also elevated (0.26), this being mainly due to the influence of polycyclic species on the overall statistics. Accordingly, the RMSE of cyclic species collectively (0.34) is substantially higher than that for acyclic species (0.15), corresponding to relative errors for the calculated $k_{298\,K}$ of about 120 % and 40 %, respectively. The large errors for cyclic species result from the difficulties in accounting fully for ring-strain and steric effects in polycyclic alkenes and cyclic poly-alkenes, as also illustrated in Fig. 2. Finally, for the full database, the SARs give fairly reliable $k_{298K}$ estimates, with a MAE of 0.13 and a RMSE of 0.21, corresponding to an overall agreement of the calculated $k_{298K}$ within about 60 %. Although this level of agreement is considered reasonable, it is noted that the methods generally do not perform as well as those for the reactions of OH with alkenes and unsaturated organic oxygenates (Jenkin et al., 2018a). This may be explained by the $O_3$ reaction being a concerted process, which is more influenced by orientational effects, ring strain and steric hindrance than the OH reaction (e.g. see Johnson et al., 2000), and therefore less easy to represent with a practical SAR. As discussed in Sects. 3 and 4, and highlighted by Vereecken et al. (2018), additional kinetics studies would be highly valuable for some classes of alkene and unsaturated oxygenate to help the SAR methods to be further assessed and refined, including data for multifunctional oxygenated species in particular.

*Data availability.* All relevant data and supporting information have been provided in the Supplement.



*Author contributions.* All authors defined the scope of the work. MEJ developed and extended the SAR methods and drafted
the manuscript, which were both reviewed by all co-authors. RV and BA tested the SAR methods in GECKO-A and carried
out the statistical analysis in Sect. 6.

*Competing interests.* The authors declare that they have no conflict of interest.

*Acknowledgements.* This work was partially funded by the European Commission through EUROCHAMP-2020 (grant
number 730997). It was performed as part of the MAGNIFY project, with funding from the UK Natural Environment
Research Council (NERC) via grant NE/M013448/1, and the French National Research Agency (ANR) under project ANR-
14-CE01-0010. It was also partially funded by the UK National Centre for Atmospheric Science (NCAS) Composition
Directorate.

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



**Table 1. Arrhenius parameters ($k = A$ exp(-($E/R$)/$T$) for the reference rate coefficients for $O_3$ addition to generic monoalkenes containing linear alkyl groups (i.e. R can be -$CH_3$ or -$CH_2R'$); and the rate coefficient values at 298 K.[a]**

| Alkene structure | Parameter | $A$ $10^{-15}$ cm$^3$ molecule$^{-1}$ s$^{-1}$ | $E/R$ K | $k_{298\,K}$ $10^{-17}$ cm$^3$ molecule$^{-1}$ s$^{-1}$ | Comment |
|---|---|---|---|---|---|
| $CH_2=CHR$ | $k_{A1O3}$ | 2.91 | 1690 | 1.0 | (b) |
| $CH_2=CR_2$ | $k_{A2O3}$ | 4.00 | 1685 | 1.4 | (c) |
| *cis*-CHR=CHR | $k_{A3O3}$ | 3.39 | 995 | 12 | (d) |
| *trans*-CHR=CHR | $k_{A4O3}$ | 7.29 | 1120 | 17 | (e) |
| CHR=$CR_2$ | $k_{A5O3}$ | 7.61 | 830 | 47 | (f) |
| $CR_2=CR_2$ | $k_{A6O3}$ | 3.00 | 300 | 110 | (g) |

Comments

(a) $k_{298K}$ and $E/R$ based on rounded averages for the sets of compounds identified in subsequent comments, with $A = k_{298K}$/exp(-($E/R$)/298). Rate coefficients are also default values for alkenes possessing remote substituents (i.e. β or higher); (b) $k_{298K}$ based on preferred data for propene, but-1-ene, pent-1-ene, hex-1-ene, hept-1-ene, oct-1-ene and dec-1-ene; $E/R$ based on preferred data for propene, but-1-ene, pent-1-ene, hex-1-ene and hept-1-ene; (c) $k_{298K}$ based on preferred data for 2-methyl-propene, 2-methyl-but-1-ene, 2-methyl-pent-1-ene, 2-methyl-hept-1-ene, 2-methyl-oct-1-ene, 2-methyl-dec-1-ene and 2-methyl-undec-1-ene; $E/R$ based on preferred data for 2-methyl-propene and 2-methyl-but-1-ene; (d) $k_{298K}$ based on preferred data for *cis*-but-2-ene, *cis*-pent-2-ene, *cis*-hex-2-ene, *cis*-hex-3-ene, *cis*-oct-4-ene and *cis*-dec-5-ene; $E/R$ based on preferred data for *cis*-but-2-ene, *cis*-pent-2-ene and *cis*-hex-2-ene; (e) $k_{298K}$ based on preferred data for *trans*-but-2-ene, *trans*-pent-2-ene, *trans*-hex-2-ene, *trans*-hex-3-ene and *trans*-oct-4-ene; $E/R$ based on preferred data for *trans*-but-2-ene, *trans*-pent-2-ene and *trans*-hex-2-ene; (f) $k_{298K}$ based on preferred data for 2-methyl-but-2-ene, *cis*-3-methyl-pent-2-ene and *trans*-3-methyl-pent-2-ene; $E/R$ based on preferred data for 2-methyl-but-2-ene; where applicable, parameters are assumed to apply to both *cis*- and *trans*- isomers; (g) $k_{298K}$ and $E/R$ based on preferred data for 2,3-dimethyl-but-2-ene; $k_{298K}$ is also consistent with reported lower limit $k$ for 3,4-dimethyl-hex-3-ene (*cis*- and *trans*-); where applicable, parameters are assumed to apply to both *cis*- and *trans*- isomers.



**Table 2. Substituent factors $F_{\alpha\,(298)}$ (X), describing the effect of the given substituent at the α carbon atom in R groups in alkenes and α,β-unsaturated oxygenates at 298 K [a].**

| X | $F_{\alpha\,(298)}$ (X) | Comment |
|---|---|---|
| alkyl (acyclic) | 0.54 | (b) |
| -OH | 1.4 | (c) |
| -OR | 0.6 | (d) |
| -C(=O)-, [-C(=O)O-] | 0.32 | (e) |
| -OC(=O)- | 0.25 | (f) |
| -ONO$_2$ | 0.044 | (g) |

Comments

(a) The temperature dependence is assumed to be described by $F_{\alpha}$(X) = exp(298 × ln($F_{\alpha\,(298)}$(X))/$T$); (b) Based on data for 3-methyl-but-1-ene, 3-methyl-pent-1-ene, *trans*-2,5-dimethyl-hex-3-ene, 2,3-dimethyl-but-1-ene and 3-methyl-2-isopropyl-but-1-ene, 3,3-dimethyl-but-1-ene, trans-2,2-dimethyl-hex-3-ene, 2.3.3-trimethyl-but-1-ene and 2.4.4-trimethyl-pent-2-ene. Also applied to organic groups containing remote substituents. The definition "acyclic" here is taken to mean that the first carbon atom in the substituent is not part of a cycle; (c) Based on data for 12 C$_3$-C$_{15}$ acyclic α,β-unsaturated alcohols; (d) Based on data for 3-ethoxypropene, 3-methyl-3,4-epoxy-1-butene and 5-methyl-5-vinyl tetrahydrofuranol; (e) Based on data for 3(*Z*-)4-methylhex-3,5-dienal, 3(*E*-)4-methylhex-3,5-dienal and 4-methyl-cyclohex-3-ene-1-one. Also applied to -C(=O)O- substituents in the absence of data; (f) Based on data for allyl acetate; (g) Based on data for 2-methyl-4-nitrooxy-but-2-en-1-ol, and consistent with lower limit value for 3-methyl-2-nitrooxy-but-3-en-1-ol.




**Table 3. Optimized ring factors at 298 K, $F_{\text{ring (298)}}$, for the reactions of $O_3$ with cyclic monoalkenes, and their temperature dependences described by $F_{\text{ring}} = A_{\text{F(ring)}} \exp (-B_{\text{F(ring)}}/T)$.**

| ring | $A_{F\text{(ring)}}$ | $B_{F\text{(ring)}}$/K | $F_{\text{ring (298)}}$ | Comment |
|---|---|---|---|---|
| 5-member ring | 0.448 | -645 | 3.9 | (a) |
| 6-member ring | 0.658 | 70 | 0.52 | (b) |
| 7-member ring | 0.374 | -500 | 2.0 | (c) |
| 8-member ring | 0.204 | -780 | 2.8 | (d) |
| 10-member ring | 0.319 | 85 | 0.24 | (e),(f) |

Comments

(a) Based on data for cyclopentene, 1-methylcyclopentene and 3-methylcyclopentene ; (b) based on data for cyclohexene, 1-methylcyclohexene, 3-methylcyclohexene and 4-methylcyclohexene; (c) Based on data for cycloheptene and 1-methylcycloheptene; (d) Based on data for *cis*-cyclooctene; (e) Based on data for *cis*-cyclodecene; (f) Tentative values of $F_{\text{ring (298)}}$ of 2.1 and 12 can be derived for 9- and 11-member rings, respectively, based on limited data for structurally-complex sesquiterpenes (see Sect. 3.2). These can be applied with an approximate average value of $A_{F\text{(ring)}} = 0.3$, and $B_{F\text{(ring)}}$ values of -580 K and -1100 K, respectively.




**Table 4. Arrhenius parameters ($k = A \exp(-(E/R)/T)$) for the rate coefficients for O$_3$ addition to generic conjugated dialkene structures; and the rate coefficient values at 298 K.[a]**

| Dialkene structure | Parameter | $A$ $10^{-15}$ cm$^3$ molecule$^{-1}$ s$^{-1}$ | $E/R$ K | $k_{298\,K}$ $10^{-17}$ cm$^3$ molecule$^{-1}$ s$^{-1}$ | Comment |
|---|---|---|---|---|---|
| **CH$_2$=C(R)CH=CH$_2$** | $k_{D1O3}$ | 10 | 1980 | 1.3 | (b) |
| **CH$_2$=C(R)C(R)=CH$_2$** | $k_{D2O3}$ | 10 | 1774 | 2.6 | (c) |
| **CH$_2$=CHCH=CHR** | $k_{D3O3}$ | 10 | 1677 | 3.6 | (d) |
| **CH$_2$=C(R)CH=CHR** CH$_2$=CHC(R)=CHR CH$_2$=CHCH=CR$_2$ | $k_{D4O3}$ | 10 | 1439 | 8.0 | (e) |
| CH$_2$=C(R)C(R)=CHR CH$_2$=C(R)CH=CR$_2$ CH$_2$=CHC(R)=CR$_2$ | $k_{D5O3}$ | 10 | 1214 | 17 | (f) |
| CH$_2$=C(R)C(R)=CR$_2$ | $k_{D6O3}$ | 10 | 887 | 50 | (g) |
| **CHR=CHCH=CHR** | $k_{D7O3}$ | 10 | 1008 | 34 | (h) |
| CHR=CHCH=CR$_2$ CHR=C(R)CH=CHR | $k_{D8O3}$ | 10 | 686 | 100 | (i) |
| **CR$_2$=CHCH=CR$_2$** CHR=C(R)C(R)=CHR CHR=C(R)CH=CR$_2$ CHR=CHC(R)=CR$_2$ | $k_{D9O3}$ | 10 | 359 | 300 | (j) |
| CR$_2$=C(R)CH=CR$_2$ | $k_{D10O3}$ | 10 | 142 | 620 | (k) |
| CR$_2$=C(R)C(R)=CR$_2$ | $k_{D11O3}$ | 10 | 0 | 1000 | (l) |

Comments

(a) $k_{298K}$ for bold structures based on data for the compounds identified in subsequent comments, with other values based on trends in the data. A value of $A = 10^{-14}$ cm$^3$ molecule$^{-1}$ s$^{-1}$ adopted in all cases, based on the reported parameters for buta-1,3-diene, *trans*-penta-1,3-diene, isoprene (2-methyl-buta-1,3-diene) and 2,3-dimethyl-buta-1,3-diene. $E/R = 298 \times \ln(A/k_{298})$. Rate coefficients are also default values for conjugated dialkenes possessing remote substituents (i.e. β or higher); (b) $k_{298K}$ based on data for isoprene (2-methyl-buta-1,3-diene); (c) $k_{298K}$ based on data 2,3-dimethyl-buta-1,3-diene; (d) $k_{298K}$ based on rounded average of data for *cis*-penta-1,3-diene and *trans*-penta-1,3-diene; parameters are assumed to apply to both *cis*- and *trans*- isomers; (e) $k_{298K}$ based on data for 2-methyl-penta-1,3-diene. The same value of $k_{298K}$ is also adopted for CH$_2$=CHC(R)=CHR and CH$_2$=CHCH=CR$_2$; (f) $k_{298K}$ for CH$_2$=C(R)C(R)=CHR taken to be a factor of 2.1 greater than that for CH$_2$=CHC(R)=CHR, based on the trend in $k$ observed on going from buta-1,3-diene (0.63), to CH$_2$=C(R)CH=CH$_2$ and CH$_2$=C(R)C(R)=CH$_2$; and from CH$_2$=CHCH=CHR to CH$_2$=C(R)CH=CHR. The same value of $k_{298K}$ is also adopted for CH$_2$=C(R)CH=CR$_2$ and CH$_2$=CHC(R)=CR$_2$; (g) $k_{298K}$ for CH$_2$=C(R)C(R)=CR$_2$ taken to be a factor of 3 greater than that for CH$_2$=C(R)C(R)=CHR, based on the trend in $k$ observed on going from CHR=CHCH=CHR to CR$_2$=CHCH=CR$_2$; (h) $k_{298K}$ based on average of data for *cis*-,*trans*-hexa-2,4-diene and *trans*-,*trans*-hexa-2,4-diene; parameters are assumed to apply to all *cis*- and *trans*- isomer combinations; (i) $k_{298K}$ for CHR=CHCH=CR$_2$ taken to be a factor of 3 lower than that for CR$_2$=CHCH=CR$_2$, based on the trend in $k$ observed on going from CHR=CHCH=CHR to CR$_2$=CHCH=CR$_2$; the same value of $k_{298K}$ is also adopted for CHR=CHC(R)=CHR; (j) $k_{298K}$ based on data for 2,5-dimethyl-hexa-2,4-diene; the same value of $k_{298K}$ is also adopted for CHR=C(R)C(R)=CHR, CHR=C(R)CH=CR$_2$ and CHR=CHC(R)=CR$_2$; (k) $k_{298K}$ for CR$_2$=C(R)CH=CR$_2$ taken to be a factor of 2.1 greater than that for CR$_2$=CHCH=CR$_2$, based on the trend in $k$ observed on going from buta-1,3-diene (0.63), to CH$_2$=C(R)CH=CH$_2$ and CH$_2$=C(R)C(R)=CH$_2$; and from CH$_2$=CHCH=CHR to CH$_2$=C(R)CH=CHR; (l) $k_{298K}$ for CR$_2$=C(R)C(R)=CR$_2$ assigned the same value as $A$, this being compatible with expected increase in $k_{298K}$ relative to that for CR$_2$=CHCH=CR$_2$.




**Table 5. Optimized ring factors at 298 K, $F'_{ring\ (298)}$, for the reactions of O$_3$ with cyclic conjugated dialkenes [a].**

| ring | $F'_{ring\ (298)}$ | Comment |
|---|---|---|
| 6-member ring | 4.5 | (b) |
| 7-member ring | 0.44 | (c) |
| 8-member ring | 0.06 | (d) |

Comments
(a) These factors apply to conjugated dialkene systems that are completely within the given ring structure. In cases where the conjugated dialkene is only partially within the ring (e.g. as in the case of β-phellandrene), the appropriate value of $F_{ring}$ given in Table 3 should be applied. In the absence of data, the temperature dependence is assumed to be described by $F'_{ring} = \exp(298 \times \ln(F'_{ring\ (298)})/T)$; (b) Based on data for cyclohexa-1,3-diene, 5-isopropyl-2-methyl-cyclohexa-1,3-diene (α-phellandrene) and 1-isopropyl-4-methyl-cyclohexa-1,3-diene (α-terpinene); (c) Based on data for cyclohepta-1,3-diene; (d) Based on data for based on data for *cis-*,*cis*-cycloocta-1,3-diene ;



**Table 6. Reference rate coefficients for O$_3$ addition to acyclic vinyl aldehyde and ketone structures.[a]**

| Oxygenate structure | Parameter | $k^{\bullet}_{298\,K}$ $10^{-17}$ cm$^3$ molecule$^{-1}$ s$^{-1}$ | Comment |
|---|---|---|---|
| *Enals* | | | |
| CH$_2$=C(R)C(=O)H | $k_{VC1O3}$ | 0.12 | (b) |
| CHR=CHC(=O)H | $k_{VC2O3}$ | 0.14 | (c) |
| CR$_2$=CHC(=O)H | $k_{VC3O3}$ | 0.18 | (d) |
| CHR=C(R)C(=O)H CR$_2$=C(R)C(=O)H | $k_{VC4O3}$ | 0.57 | (e) |
| *Enones* | | | |
| CH$_2$=CHC(=O)R | $k_{VC5O3}$ | 0.52 | (f) |
| CH$_2$=C(R)C(=O)R | $k_{VC6O3}$ | 1.2 | (g) |
| CHR=CHC(=O)R CHR=C(R)C(=O)R | $k_{VC7O3}$ | 3.9 | (h) |
| CR$_2$=CHC(=O)R [CR$_2$=C(R)C(=O)R] | $k_{VC8O3}$ | 0.83 | (i) |
| *Enediones and Enonals* | | | |
| -C(=O)C(-)=C(-)C(=O)- (unspecified group can be R or H) | $k_{VC9O3}$ | 0.5 | (j) |

Comments

(a) Determined from data for the compound or sets of compounds identified in subsequent comments. Rate coefficients are also default values for related compounds possessing other remote oxygenated substituents. The values of $k^{\circ}_{298\,K}$ should be used in Eq. (5), with the globally-optimized value of $\alpha_S = 0.19$; (b) Based on preferred data for methacrolein; (c) Based on data for *trans*-but-2-enal, *trans*-pent-2-enal, *trans*-hex-2-enal, *trans*-hept-2-enal, *trans*-oct-2-enal and *trans*-non-2-enal; (d) Based on data for 3-methyl-2-butenal; (e) Based on data for *trans*-2-methyl but-2-enal and 2-methyl pent-2-enal; (f) Based on data for methyl vinyl ketone and pent-1-en-3-one; (g) Based on data for 3-methyl-3-buten-2-one; (h) Based on data for pent-3-en-2-one, hex-4-en-3-one and 3-methyl-pent-3-en-2-one; (i) Based on data for 4-methyl-pent-3-en-2-one. Also inferred to apply to CR$_2$=C(R)C(=O)R; (j) Based on data for 4-oxo-pent-2-enal and *cis*- and *trans*-3-hexen-2,5-dione. Value assumed to apply to all α,β-unsaturated keto-aldehydes, dialdehydes and diketones, except for the unique case when all unspecified groups are H, i.e. butenedial. Data for 3,4-dihydroxy-3-hexene-2,5-dione suggests that the presence of hydroxy groups has a limited effect.






**480    Table 7. Reference rate coefficients for O₃ addition to acyclic vinylic esters and acids.[a]**

| Oxygenate structure | Parameter | $k^{\bullet}_{298\,K}$ $10^{-17}$ cm³ molecule⁻¹ s⁻¹ | Comment |
|---|---|---|---|
| *Alk-1-enoic alkyl esters* | | | |
| ROC(=O)CH=CH₂ | $k_{VE1O3}$ | 0.15 | (b) |
| ROC(=O)C(R)=CH₂ | $k_{VE2O3}$ | 0.65 | (c) |
| ROC(=O)CH=CHR | | | |
| ROC(=O)CH=CR₂ | | | |
| [ROC(=O)C(R)=CHR] | | | |
| [ROC(=O)C(R)=CR₂] | | | |
| *Alkanoic alk-1-enyl esters* | | | |
| RC(=O)OCH=CH₂ | $k_{VE3O3}$ | 0.32 | (d) |
| RC(=O)OCH=CR₂ | | | |
| [RC(=O)OCH=CHR] | | | |
| RC(=O)OC(R)=CH₂ | $k_{VE4O3}$ | 0.054 | (e) |
| [RC(=O)OC(R)=CHR] | | | |
| [RC(=O)OC(R)=CR₂] | | | |
| *Alk-1-enoic acids* | | | |
| CH₂=C(R)C(=O)OH | $k_{VA1O3}$ | 0.23 | (f) |
| CHR=CHC(=O)OH | | | |
| [CHR=C(R)C(=O)OH] | | | |
| [CR₂=C(R)C(=O)OH] | | | |

Comments

(a) Determined from data for the compound or sets of compounds identified in subsequent comments. Rate coefficients are also default values for related compounds possessing other remote oxygenated substituents. Data for peroxymethacryloyl nitrate (MPAN) suggest that parameters for alk-1-enoic alkyl esters can reasonably be applied to corresponding unsaturated PANs, and by inference peracids. The values of $k^{\circ}_{298\,K}$ should be used in Eq. (5), with the globally-optimized value of $\alpha_S = 0.19$; (b) Based on preferred data for methyl and *n*-butyl acrylate; (c) Based on data for methyl, ethyl, *n*-propyl, *i*-propyl, *n*-butyl and *i*-butyl methacrylate, ethyl crotonate and ethyl 3,3-dimethyl acrylate. Also inferred to apply to ROC(=O)C(R)=CHR and ROC(=O)C(R)=CR₂; (d) Based on data for vinyl acetate, vinyl propionate and 2-methylpropenyl acetate. Also inferred to apply to RC(=O)OCH=CHR; (e) Based on data for *i*-propenyl acetate. Also inferred to apply to RC(=O)OC(R)=CHR and RC(=O)OC(R)=CR₂; (f) Based on data for methacrylic acid and *trans*-pent-2-enoic acid. Also inferred to apply to bracketed structures shown.



**Table 8. Reference rate coefficients for O$_3$ addition to vinylic ethers.[a]**

| Oxygenate structure | Parameter | $k^{\bullet}_{298\,K}$ $10^{-17}$ cm$^3$ molecule$^{-1}$ s$^{-1}$ | Comment |
|---|---|---|---|
| ROCH=CH$_2$ | $k_{VO1O3}$ | 17 | (b) |
| ROC(R)=CH$_2$ [ROC(R)=CR$_2$] | $k_{VO2O3}$ | 1.3 | (c) |
| ROCH=CHR [ROCH=CR$_2$] | $k_{VO3O3}$ | 42 | (d) |
| (RO)$_2$C=C< [ROC(-)=C(-)OR] (unspecified groups can be R or H) | $k_{VO4O3}$ | 48 | (e) |

Comments

(a) Determined from data for the compound or sets of compounds identified in subsequent comments. Rate coefficients are also default values for related compounds possessing other remote oxygenated substituents. The values of $k^{\circ}_{298\,K}$ should be used in Eq. (5), with the globally-optimized value of $\alpha_S = 0.19$; (b) Based on preferred data for ethyl, $n$-propyl, $n$-butyl and $i$-butyl vinyl ether. Also consistent with data for ethylene glycol vinyl ether and ethylene glycol divinyl ether, but underestimates rate coefficient for $t$-butyl vinyl ether and overestimates rate coefficient for diethylene glycol divinyl ether; (c) Based on preferred data for 2-methoxypropene and 2-ethoxypropene; (d) Approximately based on lower limit rate coefficient for ethyl $n$-propenyl ether. Also inferred to apply to bracketed structures shown; (e) Based on data for 1,1-dimethoxyethene and assumed to apply to all other 1,1- and 1,2-dialkoxyalkenes.






**Figure 1. Preferred kinetic data for acyclic monoalkenes at 298 K as a function of carbon number, and the assigned rate coefficients for the generic monoalkene structures with linear alkyl substituents, as given in Table 1 (N.B. The α-branched R data in the *trans*-CHR=CHR panel consist of two co-incident data points).**





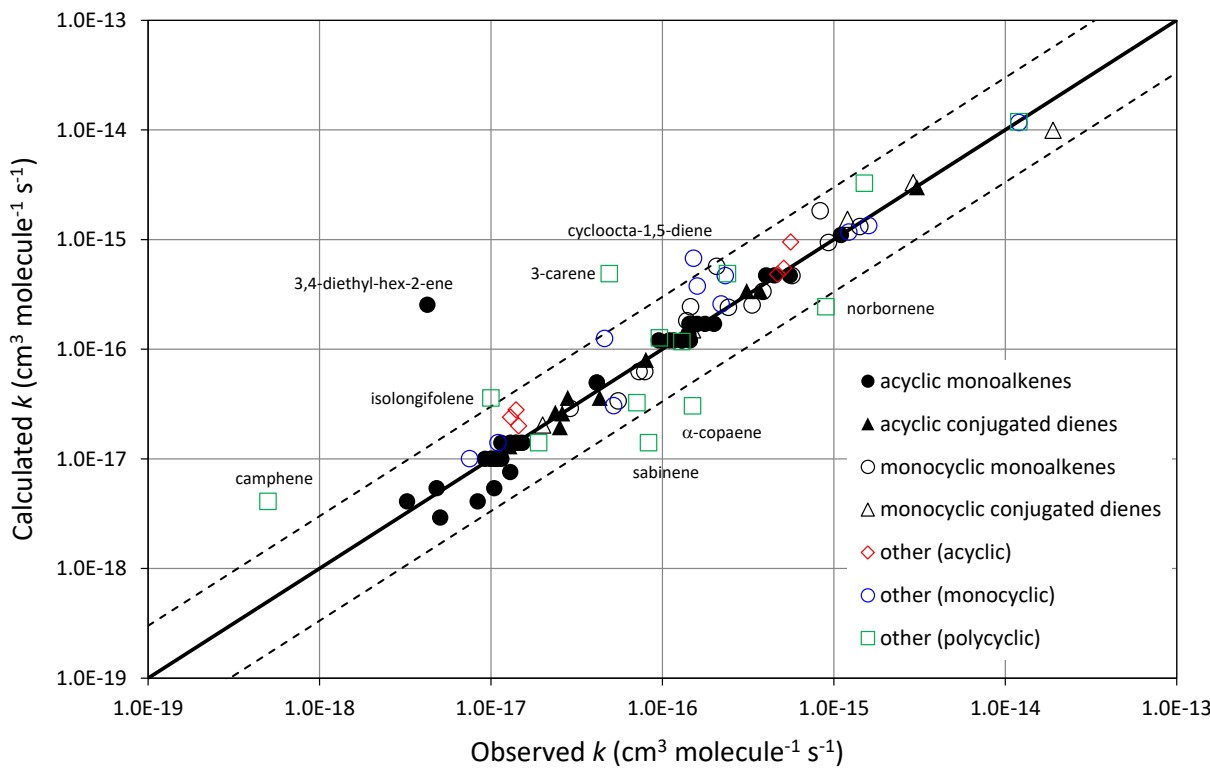

**Figure 2. Scatter plot of observed and calculated values of *k* for the reactions of O₃ with monoalkenes and poly-alkenes, based on the parameters determined in the present work. The broken lines show the factor of 3 range, within which the majority of data fall.**



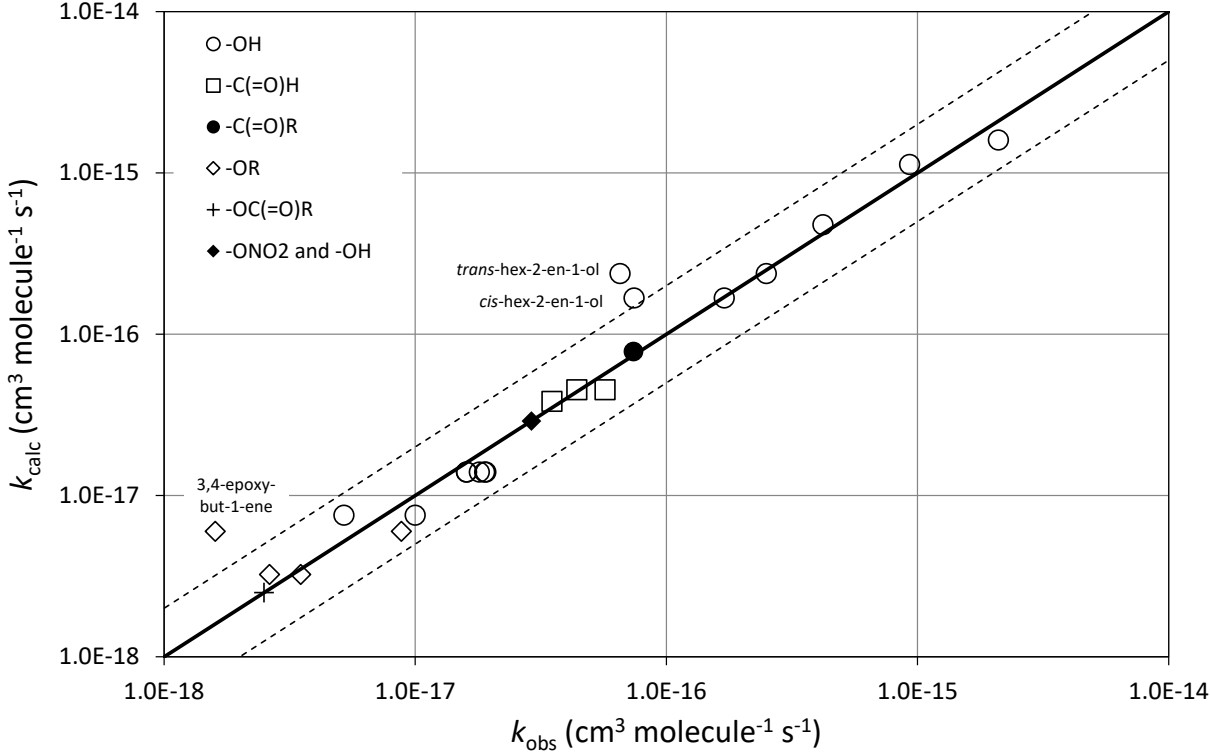

Figure 3. Scatter plot of observed and calculated values of $k$ for the reactions of O₃ with α,β-unsaturated, or allylic, oxygenates possessing the groups shown (see Sect. 4.1). The broken lines show the factor of 2 range.





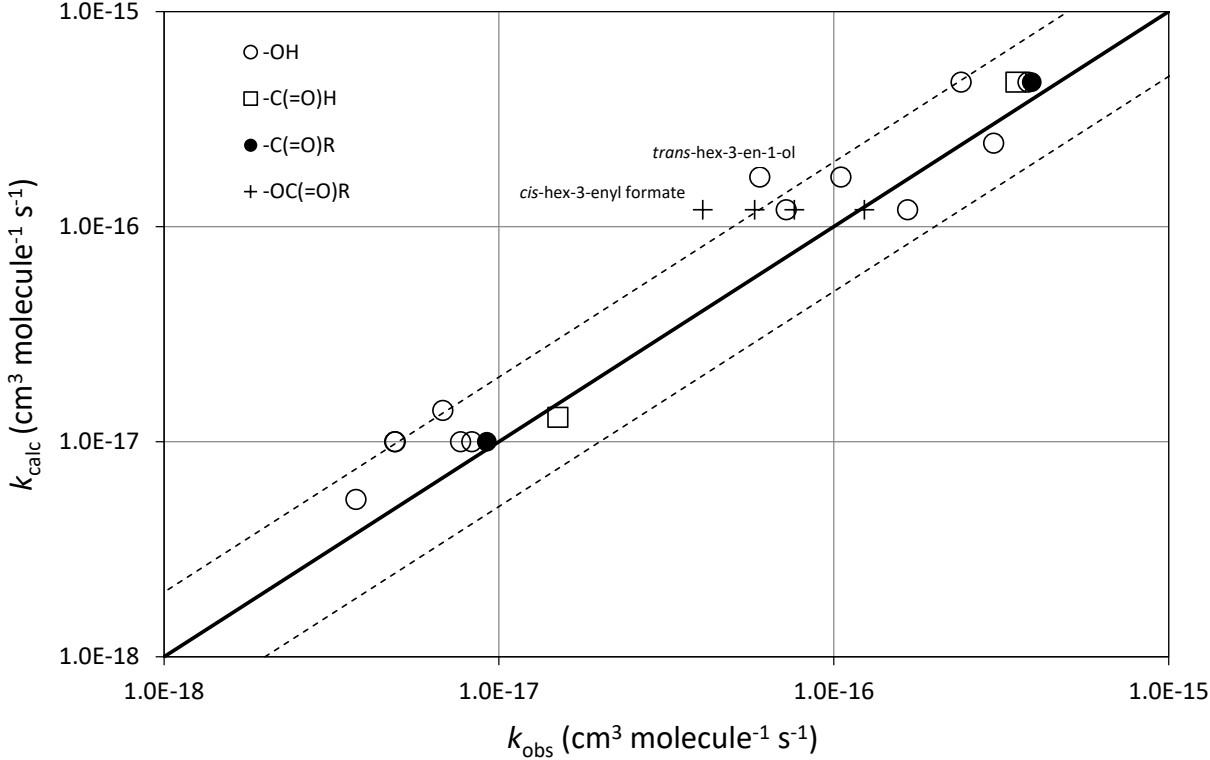

**Figure 4. Scatter plot of observed and calculated values of *k* for the reactions of O₃ with oxygenates containing remote oxygenated groups as shown (see Sect. 4.2). The broken lines show the factor of 2 range.**





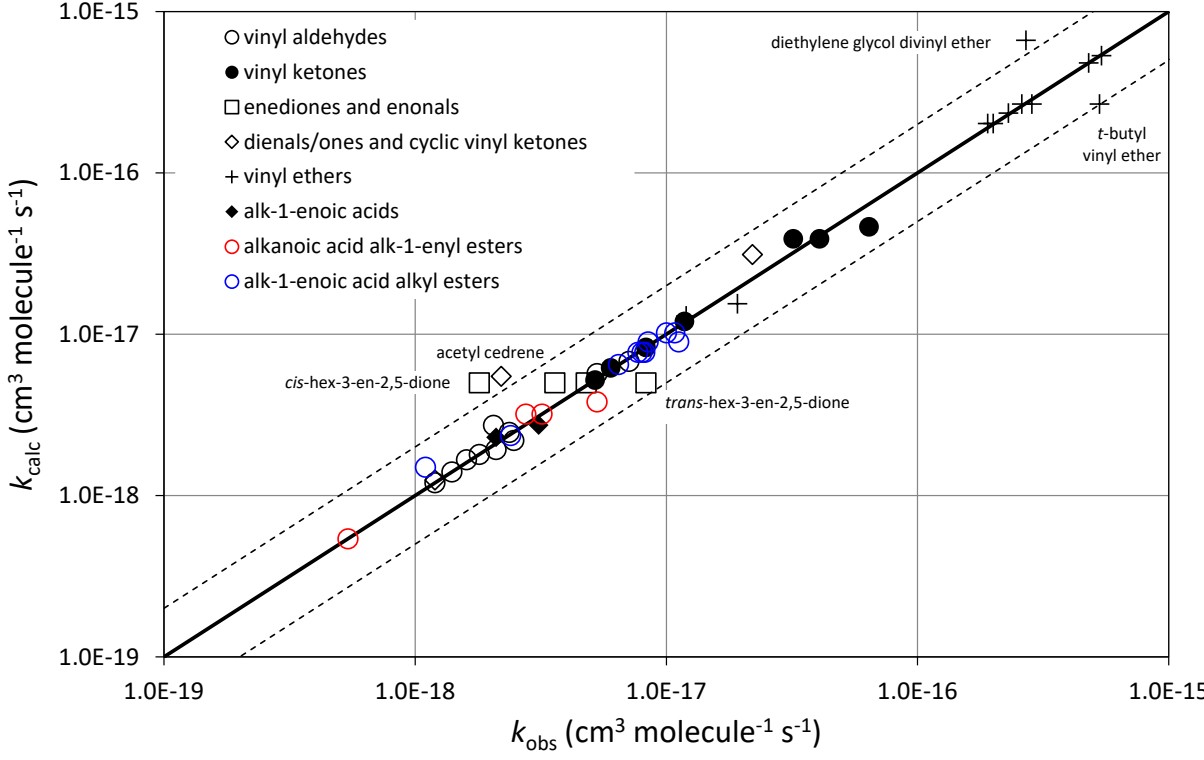

**Figure 5. Scatter plot of observed and calculated values of *k* for the reactions of O₃ with vinylic oxygenates (see Sect. 4.3). The broken lines show the factor of 2 range.**

510


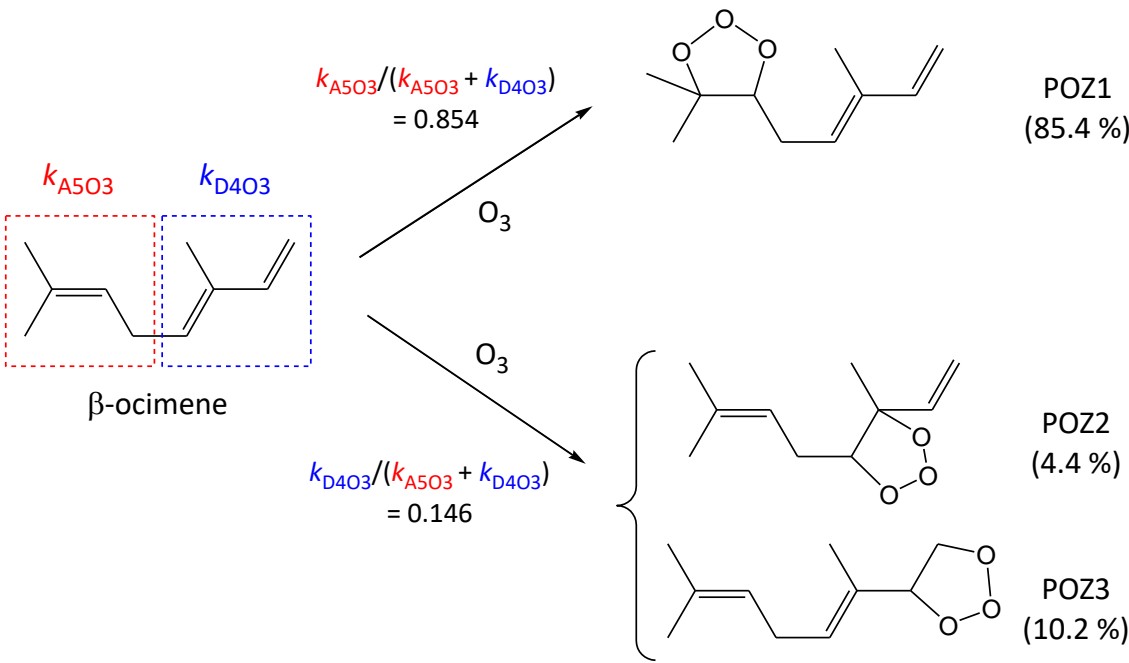

515

**Figure 6. Schematic of the concerted addition of O₃ to β-ocimene to form primary ozonides (POZ), showing the calculated contributions of the different pathways, based on the reported estimation methods (see Sect. 5).**





|  | All data | acyclic | cyclic | mono-alkene | conjug. diene | poly-alkene | allylic oxyg. | remote oxyg. | vinylic oxyg. |
|---|---|---|---|---|---|---|---|---|---|
| # species | 199 | 151 | 48 | 68 | 16 | 15 | 24 | 22 | 54 |
| RMSE | 0.21 | 0.15 | 0.34 | 0.26 | 0.11 | 0.27 | 0.18 | 0.23 | 0.14 |
| MAE | 0.13 | 0.10 | 0.23 | 0.15 | 0.07 | 0.20 | 0.12 | 0.19 | 0.08 |
| MBE | 0.04 | 0.03 | 0.07 | 0.01 | −0.02 | 0.18 | 0.00 | 0.16 | 0.03 |

**Figure 7. Root mean square error (RMSE), mean absolute error (MAE), mean bias error (MBE) and box plot for the error distribution in the estimated log $k_{298\,K}$ values for the full set and subsets of the unsaturated species in the database. The bottom and the top of the boxes are the 25th (Q1) and 75th percentiles (Q3), the black band is the median value. The whiskers extend to the most extreme data point which is no more than 1.5 × (Q3–Q1) from the box. The black dotted lines correspond to agreement within a factor 2.**