# Peer review of "Estimation of rate coefficients for the reactions of O3 with unsaturated organic compounds for use in automated mechanism construction"

_Atmospheric Chemistry and Physics, 2020_

## Referee Comment (RC1) · Anonymous Referee #1 · 7 Jul 2020

This is an outstanding paper reporting a method for automated estimation of rate co-efficients for reactions of ozone with a wide range of unsaturated organic compounds. The method described will be of great use to the atmospheric chemistry community.

---

## Referee Comment (RC2) · Anonymous Referee #2 · 31 Jul 2020

The authors present a useful and detailed structure-reactivity based system for the prediction of rate coefficients for reactions of ozone with unsaturated hydrocarbons, including oxygenated species. The new SAR, created from an up-to-date set of more than 200 measured rate coefficients, will be very useful for the development of detailed chemical mechanisms used in atmospheric modeling, and potentially other applications. The authors have done a good job of acknowledging and dealing with the many challenges associated with these ozonolysis reactions, including substituent and steric effects, molecular size, etc.

[Figure]

While I think this is clearly a publishable body of work, I do have some comments that I would like to see addressed prior to publication. These are listed below.

Line 80: Are the averages referred to here obtained from averages of A-factors and E/R values, or is a different process used? (Also, there are a couple of A-factors in the Supplementary Table for the 1-enes that look incorrect).

Line 160: While some examples are given, it is not totally clear to me what the 30 compounds referred to here are, and what distinguishes them from those discussed previously. Can more explanation be given, or can these compounds be highlighted somehow in the supplemental tables?

Line 180 or so: It was not immediately obvious to me what was being called an a,b-unsaturated compound – e.g. $CH_2=CH-CH_2-CH(=O)$ and $CH_2=CH-CH_2-OH$ are both in this category but with different numbers of C-atoms. Maybe a couple of explicit examples can be given here to clarify?

Line 206: There is a $CH_2$ group missing in the structure shown.

Line 270 or so: It seems appropriate to reference some of the work done by Moortgat and co-workers on sesquiterpene ozonolysis, where I believe some relative reactivity of different sites within a molecule are discussed/quantified.

While I realize that data are limited, and assumptions must be made, the discussion or presentation of the temperature dependence of the rate coefficients is too limited in my opinion, and often buried in figure captions, etc. I think that this information needs to be presented more fully in the main text. For example, I think the only mention of T-dependent parametrization for non-oxygenates occurs in the captions to Tables 2-5. Further discussion of temperature dependence and its uncertainties could also be given at the end of Section 3 (where a summary of non-oxygenated species is given), and/or in Section 6.

[Figure]

2020.

---

## Author Comment (AC1) · 7 Sep 2020

**Authors' responses to referee and discussion comments on:** Jenkin et al., Atmos. Chem. Phys. Discuss., https://doi.org/10.5194/acp-2020-583.

We are very grateful to the referees for their supportive comments on this work, and for helpful suggestions for modifications and improvements. Responses to the comments are provided below in Sects. A and B (the original comments are shown in blue font).

During the discussion period, we became aware that the rate coefficient for the reaction of $O_3$ with 3,4-dihydroxy-3-hexene-2,5-dione in our database is probably invalid, and this reaction has now been removed. This is discussed in Sect. C, and the resultant (minor) changes to the manuscript are described.

**A. Comments by Referee 1**

**General comment**:

This is an outstanding paper reporting a method for automated estimation of rate coefficients for reactions of ozone with a wide range of unsaturated organic compounds. The method described will be of great use to the atmospheric chemistry community.

Response: We are grateful to the referee for these very positive and supportive comments on our work.

**B. Comments by Referee 2**

**Opening comment:**

The authors present a useful and detailed structure-reactivity based system for the prediction of rate coefficients for reactions of ozone with unsaturated hydrocarbons, including oxygenated species. The new SAR, created from an up-to-date set of more than 200 measured rate coefficients, will be very useful for the development of detailed chemical mechanisms used in atmospheric modeling, and potentially other applications. The authors have done a good job of acknowledging and dealing with the many challenges associated with these ozonolysis reactions, including substituent and steric effects, molecular size, etc.

While I think this is clearly a publishable body of work, I do have some comments that I would like to see addressed prior to publication. These are listed below.

Response: We thank the referee for these positive comments on our work; and for the helpful suggestions for improvements and clarifications, which are dealt with in the responses below.

**Comment B1**: Line 80: Are the averages referred to here obtained from averages of A-factors and E/R values, or is a different process used? (Also, there are a couple of A-factors in the Supplementary Table for the 1-enes that look incorrect).

Response: Thank you very much for alerting us to the A-factor errors (for hex-1-ene and hept-1-ene) in the Supplement spreadsheet. These have now been corrected, and all entries have been double-checked.

The method and basis for averaging the kinetic parameters for acyclic monoalkenes is explicitly described in the Table 1 comments, including identification of the contributing alkenes for each parameter. The averaging method is given in comment (a), and the alkenes contributing to the average in each case are given in comments (b)--(g). Thus, the first two comments read:

> "(a) $k_{298K}$ and $E/R$ based on rounded averages for the sets of compounds identified in subsequent comments, with $A = k_{298\ K}/\exp(-(E/R)/298)$. Rate coefficients are also default values for alkenes possessing remote substituents (i.e. $\beta$ or higher);

(b) $k_{298\,K}$ based on preferred data for propene, but-1-ene, pent-1-ene, hex-1-ene, hept-1-ene, oct-1-ene and dec-1-ene; $E/R$ based on preferred data for propene, but-1-ene, pent-1-ene, hex-1-ene and hept-1-ene;"

In view of the referee's comment, we have included additional details in the text in the revised manuscript, and clarified that further information appears in the comments to Table 1. The relevant sentence has been edited as follows (new text in red font):

"The generic rate coefficients for $O_3$ addition to C=C bonds in acyclic monoalkene structures with differing extents of alkyl (R) substitution are given in Table 1 ($k_{A1O3}$-$k_{A6O3}$). These rate coefficients are based on averages of the preferred values of $k$ at 298 K, and of the preferred temperature coefficients, $E/R$, for the identified sets of alkenes (as described in detail in the Table 1 comments), and are defined for "R" being a linear alkyl group (i.e. -$CH_3$ or -$CH_2R'$)."

The referee makes a related broader point in Comment B6 (below) about lack of comment and discussion of the temperature dependences in the text. We agree that more information could be provided, and we have now included additional comment and discussion in the revised manuscript. This is described below in the response to Comment B6.

**Comment B2**: Line 160: While some examples are given, it is not totally clear to me what the 30 compounds referred to here are, and what distinguishes them from those discussed previously. Can more explanation be given, or can these compounds be highlighted somehow in the supplemental tables?

Response: The referee raises a fair point. As a result, we have added two further columns in spreadsheet SI_1 in the Supplement to help clarify the identities of all 100 alkenes within the various sets being discussed. The first additional column identifies those alkenes for which the preferred values contributed to the optimisation of the parameters presented in Tables 1–5. The second classifies the alkenes according to the categories shown in Fig. 2 and discussed in Sect. 3.5 (i.e. the section containing line 160). The second column therefore identifies the remaining 30 compounds, and classifies them as "other (acyclic)", "other (monocyclic)" or "other (polycyclic)". As indicated in the opening sentence of the Sect. 3.5, these are acyclic and cyclic compounds containing various combinations of isolated double bonds and conjugated dialkene structures, and are alkenes for which the preferred values did not contribute to parameter optimisation. The new information in spreadsheet SI_1 is also now referred to at several relevant points in Sect. 3.5 of the revised manuscript.

**Comment B3:** Line 180 or so: It was not immediately obvious to me what was being called an a,b-unsaturated compound – e.g. CH2=CH-CH2-CH(=O) and CH2=CH-CH2-OH are both in this category but with different numbers of C-atoms. Maybe a couple of explicit examples can be given here to clarify?.

Response: The category contains compounds of generic structure C=C-C-X, where "X" can be any of a number of (identified) oxygenated groups. We agree with the referee that the general use of the term "$\alpha,\beta$-unsaturated" for these compounds is confusing. This is specifically because of an ambiguity relating to carbonyl compounds, where the oxygenated substituent (i.e. C=O) contains a carbon atom.

We also describe the relevant compounds as "allylic", and now use that alternative term throughout in the revised manuscript. The relevant section (Sect. 4.1) is therefore now entitled "Allylic oxygenated compounds"; and the compounds within the category are now defined as follows in the opening sentence of the section (adjusted text in red font):

"Preferred kinetics data at 298 K are available for 25 allylic oxygenated compounds, containing the following substituents at the $\alpha$ carbon atom: -OH (14 compounds), -C(=O)H (3 compounds), -C(=O)R (1 compound), -OR (4 compounds, one possessing a remote -OH group), -OC(=O)R (1 compound), and both -$ONO_2$ and -OH (2 compounds, with one having only a lower limit recommendation)."

We feel that use of the term "allylic" clearly distinguishes these compounds from the "vinylic oxygenated compounds" (i.e. C=C-X) discussed in Sect. 4.3, where "X" is a direct substituent of the double-bonded carbon atom (i.e. as in vinyl ketones, C=C-C(=O)R, and vinyl ethers, C=C-OR, where the former also contains an additional carbon atom).

Similar to our response to Comment B2 (above), we have also now added two further columns in spreadsheet SI_2 in the Supplement to help clarify the identities of all the oxygenated compounds within the various categories being discussed. The first additional column identifies those compounds for which the preferred values contributed to the optimisation of the parameters given in Table 2 and Tables 6–8. The second classifies the oxygenated compounds according to the categories shown in Figs. 3–5 and discussed in Sects. 4.1–4.4.

**Comment B4:** Line 206: There is a CH2 group missing in the structure shown.

Response: Thank you for alerting us to this error. The structure has been corrected to "*cis*-CH₃CH₂CH=CHCH₂CH₂OC(=O)R" in the revised manuscript.

**Comment B5:** Line 270 or so: It seems appropriate to reference some of the work done by Moortgat and co-workers on sesquiterpene ozonolysis, where I believe some relative reactivity of different sites within a molecule are discussed/quantified.

Response: We agree that the work of Moortgat and co-workers has made an enormous contribution to the kinetics database for ozone-alkene reactions, and is intrinsic to many of the preferred rate coefficients and other information adopted from Calvert et al. (2000; 2011), Atkinson and Arey (2003) and the IUPAC evaluation.

Specifically, we are aware of the sesquiterpene studies of Moortgat and co-workers, which were able to report rate coefficients and OH yields from the sequential ozonolysis of the double bonds in $\beta$-caryophyllene (Winterhalter et al., 2009) and $\alpha$-humulene (Beck et al., 2011), and OH yields for sequential ozonolysis of the double bonds in a number of other terpenoid species (Herrmann et al., 2010). Although this is very instructive and useful information, we feel it is not as appropriate and helpful as the carefully chosen example ($\beta$-ocimene) to provide a clear demonstration and assessment of the methods. The two main reasons are:

(i) As discussed in Sect. 3.2 and the Table 3 comments, the rate coefficients for $\beta$-caryophyllene and $\alpha$-humulene are used as the basis for tentative ring factors assigned to 9- and 11-member rings. As a result, it is less informative to compare $k_{obs}$ and $k_{calc}$ for these compounds than it is for others, such as for $\beta$-ocimene, which $k_{obs}$ did not contribute to parameter optimisation.

(ii) The rate coefficients for the sequential ozonolysis of the double bonds in $\alpha$-humulene is not a direct measure of the reactivity of the three double bonds in the parent compound, but a comparison of their composite reactivity with the composite reactivity of those in the sets of first- and second-generation products. Although this is very interesting information, it cannot be used for direct assessment of the distribution of attack using the estimation methods. In the first- and second-generation product sets, the ring in the parent compound has been opened (or modified) such that the ring-strain is either absent, or different from that in $\alpha$-humulene itself. In practice, it is expected that the three double bonds in the parent will have comparable reactivity (as discussed by Beck et al., 2011) and contribute to the rate coefficient. The lower reactivity they report for the first-generation set is consistent with the loss of ring strain, and with the large ring-factor we have tentatively assigned to 11-member rings.

From these considerations, it is apparent that using these compounds as examples would not allow a uncomplicated demonstration of the developed methods, and it is therefore not clear how to cite them in relation to this discussion.

Finally, it should be noted that the use of β-ocimene as an example also allowed demonstration and discussion of the relative attack distribution within conjugated dienes, which would not have been possible with the sesquiterpenes.

**Comment B6 (see also Comment B1):** While I realize that data are limited, and assumptions must be made, the discussion or presentation of the temperature dependence of the rate coefficients is too limited in my opinion, and often buried in figure captions, etc. I think that this information needs to be presented more fully in the main text. For example, I think the only mention of T-dependent parametrization for non-oxygenates occurs in the captions to Tables 2-5. Further discussion of temperature dependence and its uncertainties could also be given at the end of Section 3 (where a summary of non-oxygenated species is given), and/or in Section 6.

Response: When preparing the manuscript, we were careful to ensure that a representation of the temperature dependence was provided for all parameters, even though this was quite often based on very limited information and assumption. The information generally appears in the tables and/or table comments - this approach generally being preferred by those applying SARs.

However, the referee raises a valid point, and we agree that the discussion of the temperature dependence of the reactions and parameters in the main text was too limited in the originally submitted manuscript. We have therefore amended the text at a number of points to include this information and/or to refer to where the information is given in the tables. In addition to the change indicated in our response to Comment B1 (above), the changes listed below have been made in the revised manuscript (new or amended text in red font).

In Sect. 3.1 on acyclic monoalkenes (line 100 of original manuscript), the following sentence has been added:

"In the absence of reported rate coefficients as a function of temperature for α-branched alkenes, the temperature dependence of $F_\alpha$(alkyl) is assumed to be described by $F_\alpha$(alkyl) = exp(298 × ln($F_{\alpha\ (298)}$(alkyl))/$T$); and further temperature-dependent data are also required for this assumption to be fully tested."

The first paragraph of Sect. 3.2 on cyclic monoalkenes has been modified as follows:

"The set of preferred values contains data for reactions of $O_3$ with 14 simple monocyclic monoalkenes containing endocyclic double bonds, including cyclopentenes, cyclohexenes, cycloheptenes, cyclooctenes and cyclodecenes, with the temperature dependence also defined in seven cases. The values of $k$ for these sets of compounds show systematic deviations from those observed for acyclic monoalkenes with the same level of substitution, likely resulting from the effects of ring-strain (e.g. Calvert et al., 2000). Table 3 provides a series of ring factors, $F_{ring}$, based on optimization to the 298 K rate coefficients and $E/R$ values within this dataset (as described in detail in the Table 3 comments)."

The following sentences have been added at the end of Sect. 3.3 on acyclic conjugated dialkenes:

"Temperature-dependent recommendations are available for four acyclic conjugated dialkenes (see Table 4 comments), with the recommended pre-exponential factor, $A$, being close to $10^{-14}$ $cm^3$ $molecule^{-1}$ $s^{-1}$ in each case. This value of $A$ is therefore adopted for all of the generic rate coefficients $k_{D1O3}$-$k_{D11O3}$, with $E/R$ given by 298 × ln($A/k_{298\ K}$)."

In Sect. 3.4 on acyclic conjugated dialkenes (after Eq. (4)), the following sentence has been added:

In the absence of data, the temperature dependence is assumed to be described by $F'_{ring}$ = exp(298 × ln($F'_{ring\ (298)}$)/$T$).

At the end of Sect. 4.1 on allylic oxygenated compounds, the following information has been added:

"There are almost no temperature-dependent data for allylic oxygenated compounds, and the temperature dependence of $F_\alpha$(X) is therefore assumed to be described by $F_\alpha$(X) = exp(298 × ln($F_{\alpha\ (298)}$(X))/$T$). The recent study of Kalalian et al. (2020) reports temperature dependences for the reactions of $O_3$ with cis-pent-2-en-1-ol and pent-1-en-3-ol. In the former case, the reported value of $(E/R)_{obs}$

= 902 K is very well described using the above assumption, which leads to $(E/R)_{calc}$ = 895 K; whereas in the latter case the values differ by about a factor of two, $(E/R)_{obs}$ = 730 K and $(E/R)_{calc}$ = 1590 K. Clearly, further temperature-dependent data are required for a variety of allylic oxygenated compounds for the method to be fully tested and refined."

In Sect. 4.3 on vinylic oxygenated compounds (line 233 in original manuscript), the following sentence remains unchanged:

"There is only limited information available on the temperature dependences of these reactions. Where data are available (e.g. for methacrolein and methyl vinyl ketone), the data suggest that the temperature dependence can reasonably be represented by $k = A \times \exp(-(E/R)/T)$, with $A = 10^{-15}$ cm$^3$ molecule$^{-1}$ s$^{-1}$, and $E/R = 298 \times \ln(A/k_{298\,K})$, and this approach is adopted in the present work."

Finally, we reiterate our thanks to Referee 2 for providing helpful suggestions for improvements and clarifications to the manuscript.

**C. Additional changes**

While reviewing the paper, we became aware of an apparently invalid rate coefficient in our database, which was adopted from the review of Calvert et al. (2011). The rate coefficient for the reaction of $O_3$ with 3,4-dihydroxy-3-hexene-2,5-dione in Table V-E-18 of Calvert et al. (2011), attributed to Liu et al. (1999), actually appears to be their rate coefficient for the reaction of $O_3$ with 3-hexen-2,5-dione; and this is also indicated by the associated discussion in the text of Calvert et al. (2011). This apparently erroneous tabulated entry has since also appeared in Calvert et al. (2015) and McGillen et al. (2020). We are not aware of any reported rate coefficients for the reaction of $O_3$ with 3,4-dihydroxy-3-hexene-2,5-dione, and this reaction has therefore been removed from our database, which now includes data for 110 unsaturated oxygenated compounds. This has resulted in the following minor changes to the manuscript:

(i) References to the numbers of reactions in the database and in the statistical analysis have been adjusted accordingly in the Abstract and Sects. 1, 2 and 7.

(ii) The brief discussion of 3,4-dihydroxy-3-hexene-2,5-dione as the sole compound containing vinylic hydroxy groups in the database (previously at the end of Sect. 4.3) has been revised as follows, now referring to data for pentane-2,4-dione (acetyl acetone) and its enolic tautomer (new and revised text in red font):

"There are no data for compounds containing a number of vinylic oxygenated functional groups (e.g., -ONO$_2$, -OOH), although such compounds are not expected to be prevalent in atmospheric chemistry. Data for the reactions of $O_3$ with vinylic alcohols are very limited because kinetics studies tend to be complicated by keto/enol tautomerism. A recent theoretical study of the reaction of $O_3$ with 4-hydroxy-pent-3-en-2-one (Ji et al., 2020), the enolic tautomer of pentane-2,4-dione (acetyl acetone), suggests that the presence of the hydroxy group has a limited effect, the reported rate coefficient at 298 K (2.4 × 10$^{-17}$ cm$^3$ molecule$^{-1}$ s$^{-1}$) being comparable with $k_{VC7O3}$. However, it is noted that this is more than an order of magnitude greater than the laboratory determination of Zhou et al. (2008) for the reaction of $O_3$ with the tautomeric mixture of pentane-2,4-dione and 4-hydroxy-pent-3-en-2-one. Further information is clearly required to allow the effects of vinylic hydroxy groups to be defined with confidence. Until then, they are provisionally assumed to have the same influence as vinylic H atoms in the present work."

**References**

Beck, M., Winterhalter, R., Herrmann, F., and Moortgat, G. K.: The gas-phase ozonolysis of $\alpha$-humulene, Phys. Chem. Chem. Phys., 13, 10970-11001, 2011.

Calvert, J. G., Mellouki, A., Orlando, J. J., Pilling, M. J. and Wallington, T. J.: The mechanisms of atmospheric oxidation of the oxygenates, Oxford University Press, Oxford, ISBN 978-0-19976707-6, 2011.

Calvert, J. G., Orlando, J. J., Stockwell, W. R. and Wallington, T. J.: The mechanisms of reactions influencing atmospheric ozone, Oxford University Press, Oxford, ISBN 978-0-19023302-0, 2015.

Herrmann, F., Winterhalter, R., Moortgat, G. K. and Williams, J.: Hydroxyl radical (OH) yields from the ozonolysis of both double bonds for five monoterpenes, Atmos. Environ., 44, 3458-3464, 2010.

Ji, Y., Qin, D., Zheng, J., Shi, Q., Wang, J., Lin, Q., Chen, J., Gao, Y., Li, G. and An, T.: Mechanism of the atmospheric chemical transformation of acetylacetone and its implications in night-time second organic aerosol formation, Sci. Total Environ., 720, 137610, 2020.

Kalalian, C., El Dib, G., Singh, H. J., Rao, P. K., Roth, E. and Chakir, A.: Temperature dependent kinetic study of the gas phase reaction of ozone with 1-penten-3-ol, *cis*-2-penten-1-ol and *trans*-3-hexen-1-ol: Experimental and theoretical data, Atmos. Environ., 223, 117306, 2020.

Liu, X., Jeffries, H. E. and Sexton, K. G.: Atmospheric photochemical degradation of 1,4-unsaturated dicarbonyls, Environ. Sci. Technol., 33, 4212-4220, 1999.

McGillen, M. R., Carter, W. P. L., Mellouki, A., Orlando, J. J., Picquet-Varrault, B., and Wallington, T. J.: Database for the kinetics of the gas-phase atmospheric reactions of organic compounds, Earth Syst. Sci. Data, 12, 1203–1216, https://doi.org/10.5194/essd-12-1203-2020, 2020.

Winterhalter, R., Herrmann, F., Kanawati, B., Nguyen, T. L., Peeters, J., Vereecken, L., and Moortgat, G. K.: The gas-phase ozonolysis of $\beta$-caryophyllene ($C_{15}H_{24}$). Part I: an experimental study, Phys. Chem. Chem. Phys., 11, 4152-4172, 2009.

Zhou, S., Barnes, I., Zhu, T., Bejan, I., Albu, M. and Benter, T.: Atmospheric chemistry of acetylacetone, Environ. Sci. Technol. 42, 7905-7910, 2008.